# Nanoval Technology—An Intermediate Process between Meltblown and Spunbond

**DOI:** 10.3390/ma16072932

**Published:** 2023-04-06

**Authors:** Tim Höhnemann, Johannes Schnebele, Walter Arne, Ingo Windschiegl

**Affiliations:** 1German Institutes of Textile and Fiber Research (DITF), Koerschtalstr. 26, D-73770 Denkendorf, Germany; 2Fraunhofer Institute for Industrial Mathematics (ITWM), Fraunhofer Platz 1, D-67663 Kaiserslautern, Germany

**Keywords:** polymers, processing, fibers, spinning, meltblown, spunbond, nonwovens, numerical simulations, modeling, rheology

## Abstract

The idea of ”Nanoval technology“ origins in the metal injection molding for gas atomization of metal powders and the knowledge of spunbond technologies for the creation of thermoplastic nonwovens using the benefits of both techniques. In this study, we evaluated processing limits experimentally for the spinning of different types of polypropylene, further standard polymers, and polyphenylene sulfide, marked by defect-free fiber creation. A numerical simulation study of the turbulent air flow as well as filament motion in the process visualized that the turnover from uniaxial flow (initial stretching caused by the high air velocity directed at the spinning die) to turbulent viscoelastic behavior occurs significantly earlier than in the melt-blown process. Modeling of the whole process showed that additional guide plates below the spinneret reduce the turbulent air flow significantly by regulating the inflow of secondary process air. The corresponding melt flow index of processible polymer grades varied between 35 g·10min^−1^ up to 1200 g·10min^−1^ and thus covering the range of extrusion-type, spunbond-type, yarn-type, and meltblown-type polymers. Hence, mean fiber diameters were adjustable for PP between 0.8 and 39.3 μm without changing components of the process setup. This implies that the Nanoval process enables the flexibility to produce fiber diameters in the typical range achievable by the standard meltblown process (~1–7 μm) as well as in the coarseness of spunbond nonwovens (15–30 μm) and, moreover, operates in the gap between them.

## 1. Introduction

Both the meltblown and the spunbond processes are established industrial technologies to produce nonwoven fabrics from polymer melt. Due to the respective process layout, the resulting nonwovens show different, characteristic web properties (mainly regarding the resulting fiber diameters and thus the haptic, stiffness, and mechanical strength) and are thus predestined for different areas of application or different functions in the same application, respectively. However, both processes are comparable considering the web formation because they involve collecting fibers randomly or in the desired direction on a moving belt with a vacuum/suction box located below the belt. This is different from the defined winding of endless filaments in the melt spinning of yarns [1,2]. The first process step of meltblown and spunbond processes is similar as well, considering the feeding of the extrusion line with resin, melting, and transport under a defined flow rate (usually using a gear pump) to a spin beam, passing a spin pack with filters [1,2,3]. Crucial differences are the spinneret type itself and the way of applying primary and secondary process air to the polymer melt exiting the capillaries in the form of endless filaments in particular. Precisely for this reason, both the usable polymer grades differ severely, and the resulting purity of fibers is different, with some gaps in between.

The spunbond process, established for synthetic polymers in the 1960s by Freudenberg and Du Pont [3,4,5], is quite similar to the melt spinning of yarns, with the difference that industrial spunbond spin beams use a significantly higher number of capillaries/spinnerets (1000 to >60,000 holes per meter [6] vs. around 30 to several hundred holes per spin pack [5,7], which are aligned in a multiple-row matrix). The melt leaves the solid spin plates, or more precisely, the multiple nozzles as filaments, which enter a duct, where they are cooled down by conditioned quench air and stretched in the following by the appliance of suction or pressure air, a combination of both, or by additional support of godets [8]. This main part of the process can be considered as non-isothermal extensional flow of the melt [9], where stress-induced crystallization is applied depending on the adjacent speed [10], whereby the air speed is a factor of 1.5 to 4 higher than the filament velocity, which reaches around 6000 m·min^−1^ at its maximum [1]. The focus of the spunbond process is more on the strength of the fibers than their purity, obviously.

In the meltblown process, the melt stream flows out of a prismatic linear assembly of several hundred die orifices, usually [5], and gets captured by two convergent streams of hot and high-velocity primary process air (“Exxon-type die”). This shape of the die seriously limits the selection of usable polymers due to its low pressure resistance. On the way to the conveyor belt (die-collector distance, DCD), the primary process accelerates and stretches the fibers immediately. Together with the inflow of secondary air (around four times the amount of primary air [11]) (Value generated by process-near simulations of the meltblown process by Fraunhofer ITWM, Kaiserslautern) a turbulent meltblown free jet is formed [8,12] where the fibers become swirled and deflected [5] and the final diameter of mostly fine fibers is reached. Because the drawing of the fibers takes place while they are still in the ‘’semi-molten state’’, there is no downstream drawing method before their deposition, and thus, meltblown nonwovens only exhibit low to moderate strength [5].

Both technologies lay down a stochastic nonwoven web in one process step, where the fibers are held together by a combination of entanglement and cohesive sticking [5]. Moreover, they can be further divided into different processes, but all of them are similar in their respective process principles. On the commercial side, there are the Reifenhauser Reicofil spunbonding [13], the Nordson’s MicroFilTM spun bond system [14,15], or the Docan^®^ spunbond process of Lurgi GmbH/STP Impianti (SpA process) [1] to be named as the most essential systems in spunbond. The Ason-Neumag-process [5] even incorporates the benefits of meltblown to spunbond concerning high filament velocities and temperatures in order to produce finer fiber diameters than usual [16]. For meltblown, the classical process setting (“Exxon-type”) is based on the development of Exxon Mobile Corp (Irving, TX, USA). in the 1960s. [2] In addition, the special execution (“Biax-system”) of Biax Fiber System (Greenville, WI, USA) must be considered [8,17,18].

In addition to the technical circumstances of different die shapes with corresponding limitations, the different modes of action of the applied air define different requirements for the polymer properties. So, quite high viscosities of the polymer melt are required for spunbond processes, such as the melt spinning of yarns, where high forces act on the partially cooled filaments during take-up. In the meltblown process, where process air acts immediately on the melt after the filament leaves the capillaries, a lower viscosity of the polymer melt is needed. As an established quantity, the melt flow index (MFI) can be used to estimate the viscosity range of the polymer. These differences in the properties of the polymer and the different process setups of both technologies, with their specific limits, lead to significant differences in the property range of the produced nonwovens, as shown in Table 1.

A gap between both processes can be derived from this, despite the capillary arrangement. This addresses mainly the processable viscosity range of polymer used on the raw material side and the resulting diameters achievable on the fiber side. As a consequence, the typical differences for application depend on the poor mechanical properties but high filtering efficiencies of meltblown fabrics and the high mechanical strength but also highly permeable spunbond structures [1,2,3,5]. Further, spunbond processes have not been established for high-temperature polymers, such as PPS (polyphenylene sulfide) or PEEK (polyether ether ketone) so far due to the focus of industrial plants on very high productivities of standard polymers (>1000 kg·h^−1^·m^−1^). Thus, until now, almost exclusively polyolefins, polyesters, polyamides, and TPUs (thermoplastic polyurethanes) were processed on an industrial scale [1,2,3,5]. Although the industrial meltblown market is mainly focused on polypropylene and polyester nonwovens, it has a high variety of researched polymers that can be successfully used up to the use of high temperature polymers. It was successfully shown to process PPS and also PEEK with process temperatures up to 450 °C at the DITF Denkendorf in 2013 [20], exceeding the state-of-technique limits of the conventional meltblown technique.

The Biax-meltblown-system of the Biax Fiber System (Greenville/WI, USA) aims to combine the advantages of both technologies by using a field of multiple rows of capillary lines up to around 250 hpi [20]. Every single nozzle is surrounded concentrically by a preferably ring-shaped (also square or triangular) blow nozzle to supply an individual air stream around the capillary [9,18,19]. This (Biax-) Spun-Blown^®^ nonwoven process is used to “find a way to extrude smaller fibers, having a diameter close to those of meltblown fibers, yet having similar strength of spunbond fibers” [21]. In fact, spunblown polypropylene nonwovens are reported to show higher tensile strength and elongation than meltblown nonwovens and lower strength but similar elongation to spunbond nonwovens (at the same base weight) [19,22,23]. By this means, the process “consumes” a lower total amount of process air than meltblown and thus shows lower energy consumption at higher productivity but a lower stretching of the fibers compared to the spunbond process [24].

The “Nanoval technology“ [25,26,27] combines elements of the “Biax fiberfilm die” and of the “metal injection molding technology” [28] for the gas atomizing technology of metal powders [29]. In the latter, a melt passes a so-called “Laval-Nozzle” (converging-diverging nozzle) together with a process gas, which is accelerated over a short distance towards the narrowest cross section of the nozzle. The gas flow is laminar and parallel to the melt stream, while the melt is co-accelerated due to the impulse exchange of air and melt by shear forces. [30] After the narrowest point of the cross-section, the gas decompresses and accelerates further. [28,31] Here supersonic velocities are possible, dependent on the pressure ratio before and at the narrowest point of the cross-section, with the critical pressure ratio = 1.8 [28,31].

Powders are formed out of the metal melt by bursting as the surface tension overcomes the viscous forces (using a single Laval die) [31]. Transferred to polymer processing, the polymer is fed, molten, and homogenized by a common extrusion system and conducted to the spinneret with the melt distribution and capillaries arranged in a straight line and rows (at least one) [31]. The uniqueness of polymer processing is that the melt exits the spinneret together with the process air. Both hit inside the spinneret in (converging) chambers configured as Laval nozzles, which are confined by the air duct (concentric around the capillary) from the sides, the tip of the spinning capillary from above, and the spinneret hole from below [27,28,31,32,33].

The narrowest cross section is located beneath the point where the spinning material exists “its” capillary [28]. The melt becomes stretched by the shear stresses transmitted by the air stream. In this way, spunbonded nonwovens can be produced from endless threads [26]. Since every spinning hole has its own air supply, a good relationship of air distribution to the single filament is ensured [25], and a very effective impulse exchange between air and melt is possible by the Laval die [30]. The co-current acceleration that is steadily ongoing from the narrowest point is unique to the meltblown process, where the air has its maximal velocity at the air slit and decreases continually after the contact with the melt stream due to the impulse transfer. Due to the efficient impulse transfer, low air flow rates through the individual, circular Laval nozzles are sufficient, such as in the meltblown process with a long gap. In the diameter range below 3 μm finer fibers are reported to be producible at higher throughputs (3–20 g·ho^−1^·min^−1^ [34]) at a lower number of spinholes compared to meltblown (up to 10× higher [34]).

In this study, we evaluate the process limits of the Nanoval system and try to elaborate on the variability of this process to act as a multifunctional option for and in between the well-known meltspun technologies. Therefore, on the one hand, the requirements for the applied polymers and suitable process windows are characterized. On the other hand, the limits of the process are compared to the known meltspun technologies (spunbond, meltblown) as well as the achievable fiber diameters thereof. Further, the process is evaluated regarding the use of high-temperature (HT) polymers. For the simulation of the Nanoval process, a multiscale Computation Fluid Dynamics (CFD) simulation of the air flow is established. There are two different kinds of air supplies for the nozzles, which are simulated separately. The air flow conditions at the position where the melt enters the air stream serve as boundary conditions for a larger simulation of the fiber-forming region. The periodicity of the nozzle positions is utilized in this simulation to save a huge amount of computational power. The result of the CFD simulation is then used to compute the fiber dynamics. The challenge for the model of the fiber dynamics is elongation rates of several hundred thousand. To overcome this challenge, the fiber-forming process is separated into two parts. For the first millimeters after the nozzle, a uniaxial viscous model can be used, which then transitions into an unsteady viscoelastic model taking the turbulence effects into account. With this solution strategy, it is possible to monitor the fiber diameter development from the nozzle to the end diameter and all relevant aerodynamic effects such as the interaction of the free jets, the generation of the vorticity, and the behavior of the suction box.

## 2. Materials and Methods

### 2.1. Nanoval Spinning Set-Up

The setup of the Nanoval process consists of four main components, as shown in the scheme in Figure 1: the Nanoval spinning beam itself (A), an extrusion system (B), a system for the supply of the hot process air (C), and the deposition and air treatment system (D).

A single screw extruder (3-zone screw, Ø 35 mm × 30 D) from Extrudex GmbH (Mühlacker, Germany) and a gear pump from Mahr Metering Systems GmbH (Göttingen, Germany) with a volume of 10 cm^3^·1rpm^−1^ are used to melt and transport the polymer to the spinning beam. The air system consists of a compressor (Aertronic D12H) of Aerzener Maschinenfabrik GmbH (Aerzen, Germany) with an air volume flow limit of 110 Nm^3.^h^−1^ (minimal) and 440 Nm^3.^h^−1^ (maximal), combined with a flow heating system of Schniewindt GmbH & Co KG (Neuenrade, Germany), and an air distribution unit, splitting up the air supply to four ducts entering the spinning beam of Nanoval GmbH (Berlin, Germany).

The Laval dies are formed by the air ducts in the gap (L = 2.5 mm) between the tip of the capillaries of the polymer melt (Ø 0.3 mm, L/D 8) and the opening holes of the spinning beam (Ø 2.7 mm), where melt and air exit together (scheme, see Figure 1).

In the spinning room, between the collector belt and the spinning beam, an air channel of rectangular shape (460 × 500 mm, length = 200 mm, side walls constructed as perforated plate structures (10 × 10 mm hole pattern; Qg 10-20 DIN 24041)) is placed below the spinning beam in order to ensure a better handling of secondary air, reduce the air turbulence, and increase the web homogeneity (see Section 3.4). The conveyor belt of Siebfabrik Arthur Maurer GmbH & Co KG (Mühlberg, Germany) is a steel fabric tape with clip seam and silicon edging in a total width of 0.72 m (No. 16/cm linen weave) with a warp wire of 0.22 mm diameter stainless steel (1.4404 AISI 316L) and a weft wire of 0.22 mm diameter stainless steel (1.4404 AISI 316L). It has a maximal take-up velocity of 10 m·min^−1^ and can be adjusted in height from 100 mm up to 750 mm to vary the die-collector distance (DCD). Below the belt section, where the filaments are laid down, an air-suction box (suction surface of 0.128 m^2^, 20 × 64 cm) with a maximal suction volume of 2900 Nm^3^·h^−1^ (maximum flow velocity: 11 m·s^−1^) is placed to remove the process (and secondary) air.

Variable parameters of the entire system are:Polymer throughput;Process temperature (melt);Process temperature (air);Air throughput;Die-collector distance (DCD);Distance between the die and air channel (DDAC);Collector speed.

### 2.2. Production of Nonwovens

Nonwoven production trials were performed with the presented spinning setup (Section 2.1) under variations of polymer type, polymer throughput, process temperature (melt), and air throughput. The DCD (500 mm) and the DDC (100 mm) were kept constant to minimize the experimental grid. Additionally, the process temperature of the air was fixed at 30 °C above the melt temperature, which was found to be the lowest temperature that delivers the most homogeneous air/melt flow out of the spinning beam (see Section 3.1). The collector speed was varied in accordance to produce a constant area base weight of the produced nonwovens of 45 g·m^−2^ in order to obtain comparability (without the influence of the base weight) of web properties by different process settings.

### 2.3. Materials

Six different polymer types were chosen to analyze their processability in the Nanoval process, and accordingly, different types (with different MFIs or intrinsic viscosities (η)) were obtained to reveal the influence of molar mass/rheological differences: Polypropylene (PP), Polyethylene terephthalate (PET), Polybutylene terephthalate (PBT), Polyamide 6 (PA6), Polyphenylene sulfide (PPS), and Polyether ether ketone (PEEK). All polymers used and their characteristic properties are summarized in Table 2.

### 2.4. Determination of the Moisture Content

The residual water content for all polymers (despite PP) was determined by Karl Fischer titration, which was performed at 140 °C on an “899 Coulometer” and an “885 Compact Oven SC” (both: Deutsche METROHM GmbH & Co. KG, Filderstadt, Germany). The resulting water content was <150 ppm, respectively.

### 2.5. Rheological Characterization

Shear rheological experiments in the temperature and time-sweep modes were performed on a “Physica MCR 501” rheometer (Anton Paar Group AG, Graz, Austria) in plate–plate geometry at different temperatures. Polymer granules were placed on the lower plate (25 mm in diameter), and the gap was adjusted to 1.0 mm. Afterwards, excess material was removed, and the test was performed under a nitrogen atmosphere (strain: 10%, angular frequency: 10rad·s^−1^). Temperature ramps were performed under adjustment of the gap in order to maintain a constant normal force over the measurement. The strain amplitude was proven to be in the linear viscoelastic regime by strain sweep tests at a constant angular frequency of 10 rads^−1^.

Measurements of the melt volume flow rate were executed on polypropylene samples at 230 °C with a load of 2.16 kg according to *ISO 1133* using a “Göttfert MI-B” (GÖTTFERT Werkstoff-Prüfmaschinen GmbH, Buchen, Germany). Depending on the flowability (~experiment time), 10–15 data points were taken with constant time steps of seconds and the mean value calculated. Three measurements were performed per sample.

### 2.6. Nonwoven Testing

Nonwoven properties were tested to obtain comparable characteristics on the one hand and to generate information about the nonwoven homogeneity, in particular in cross direction (CD), on the other hand. Therefore, a systematic sample extraction was carried out, as shown schematically in Figure 2.

The area base weight of nonwovens was determined by cutting out and weighing square sections of 10 × 10 cm (100 cm^2^) out of the nonwovens according to *DIN EN ISO29073-1*. To analyze the homogeneity of the nonwoven, three samples were taken across the CD and also in the MD (see squares in Figure 2). To obtain a “higher resolution” of the base weight distribution across CD, five smaller round sections of 1 cm in diameter (0.782 cm^2^) were cut out of the three samples across CD (yellow scheme in Figure 2). The coefficient of variation (cv-value) of the 15 measurement values was calculated as a robust characteristic value for the nonwoven homogeneity.

In accordance with the base-weight sampling, the air permeability was measured on the 10 × 10 cm section in accordance with DIN EN ISO 29073 T1 with a probe of 20 cm^2^ and a differential pressure of 200 Pa. On the same samples, the nonwoven thickness was measured using a test head of 1 cm^2^ and a test force of 0.2 cN·cm^−2^. Eight measurements were executed diagonally along the sample (see yellow scheme in Figure 2).

The fiber diameter distribution was determined using scanning electron microscopy (SEM). Therefore, a round sample was punched out of the nonwoven and placed on the SEM carrier, which was sputtered in argon plasma (40 sec under a vacuum of 0.1 mbar, with a distance of 35 mm, a current of 33 mA, and a voltage of 280 V) with a gold-palladium layer of 10–15 nm. Three SEM micrographs per sample were taken with a magnification of ×1000 using a “TM-1000 tabletop electron microscope” of Hitachi High-Tech Corporation (Tokyo, Japan) with an accelerating voltage of 15 kV in the “charge-up reduction mode”. The magnification was chosen to catch around 40 single fibers, and contrast and illumination were adjusted to gain an image of straight monochromic fibers in front of a dark monochrome To analyze the images, the beta software “MAVIfiber2d“ of Fraunhofer ITWM (Kaiserslautern, Germany) was used [45]. First, the images were smoothed by an algorithm and binarized by the software before a statistical analysis was performed over each fiber pixel without segmentation into individual fibers [46,47]. After merging the output of the three images, the mean and median fiber diameters as well as the standard deviation and interquartile range were given out.

### 2.7. Numerical Descriptions of the Simulation Model

A complete three-dimensional numerical simulation of meltblown processes with several hundred filaments is generally not possible due to the computational effort. However, the fiber–fiber contact and the feedback of the fibers on the flow in a meltblown process are negligible, so we only consider a one-way coupling. Because the filaments are long and very thin, we do cross-sectional averaging and look at all magnitudes along the filament curve. These quantities are speed, stress, temperature, and diameter. In this work, we neglect the surface tension of the polymer fibers (which is considered to be included in upcoming work).

In the region close to the nozzle, the high-speed air stream pulls the slowly extruded fiber jet rapidly down without any lateral bending. Here, the hot temperatures prevent fiber cool down and solidification, and the fiber jet behavior is mainly determined by the mean airflow; turbulent effects are negligible. Hence, we assume that in the nozzle region the polymer jet can be described by a steady uniaxial viscous fiber model with deterministic aerodynamic forces, given in Equations (1)–(4).
(1)ddsσ=Re3 μσu+σu−1Fr2−fairu,
(2)$$\frac{d}{\mathrm{ds}}T=-\frac{\mathrm{St}}{\epsilon }\pi d\alpha \left(T-T_{\mathrm{air}}\right),$$
(3)ddsu=Re3 μσ,
(4)$$\sigma \left(1\right)=0,\;T\left(0\right)=1,\;u\left(0\right)=1,$$

From the solution of the boundary value problem with stress, temperature, and speed (σ, T, u) over the arc length s, the initialization of the instationary visco-elastic model is conducted. Re, Fr, and St are dimensionless numbers, $f_{\mathrm{air}},{\;T}_{\mathrm{air}}$ are the air force and air temperature, and μ, d, and α are viscosity, diameter, and heat transfer coefficient, respectively (for more details, see [48]). In the region further down from the nozzle, turbulent aerodynamic fluctuations crucially affect the fiber behavior. The coupling point is the nearest point to the nozzle, where the ratio of the relative velocity and the turbulent velocity scale is below one order of magnitude. From this point, the further transient fiber behavior is described by an unsteady viscoelastic fiber model accounting for turbulent effects (see Equations (5)–(10)).
(5)$$\partial_{t}r=v,$$
(6)$$\partial_{l}r=\tau ,$$
(7)$$\partial_{t}v=\partial_{l}\left(\sigma \frac{\tau }{{\left|\left|\tau \right|\right|}^{2}}\right)+\frac{1}{{\mathrm{Fr}}^{2}}e_{g}+f_{\mathrm{air}},$$
(8)$$\partial_{t}T=-\frac{\mathrm{St}}{\epsilon }\pi d\alpha \left(T-T_{\mathrm{air}}\right)\left|\left|\tau \right|\right|,$$
(9)De∂tσ−2σ+3p∂tττ+σθ=3μ∂tτReθτ,
(10)De∂tp+p∂tττ+pθ=−μ∂tτReθτ,

The limit of Deborah number De→0 leads to the pure viscous behavior with transformation from Lagranian to the arc length discretization. The additional variables are: r fiber curve, v fiber velocity, τ the tangent, p fiber pressure, and θ relaxation time. A detailed explanation of the equations is presented in [48]. The boundary conditions are as follows in Equation (11) at the free end:(11)$$\sigma \left(0,t\right)=0,\;\;\;\;\;\;p\left(0,t\right)=0,\;\;\;\;\;\;\;\left|\left|\tau \right|\right|\left(0,t\right)=1$$
at the nozzle, as shown in Equation (12)
(12)$$r\left(l_{\mathrm{end}}\left(t\right),t\right)=r_{\mathrm{in}},\;\frac{\tau }{\left|\left|\tau \right|\right|}\left(l_{\mathrm{end}}\left(t\right),t\right)=e_{g},\;\;v\left(l_{\mathrm{end}}\left(t\right),t\right)=e_{g},\;\;T\left(l_{\mathrm{end}}\left(t\right),t\right)=1.$$
and the initial conditions for t = 0 as given in Equation (13).
(13)$$\sigma \left(l_{\mathrm{end}}\left(0\right),0\right)=\sigma_{\mathrm{in}},\;\;p\left(l_{\mathrm{end}}\left(0\right),0\right)=p_{\mathrm{in}},\;\left|\left|\tau \right|\right|\left(l_{\mathrm{end}}\left(0\right),0\right)=1,$$

The turbulence reconstruction is based on the flow variables of the kinematic turbulent energy and the dissipation. All the modeling and numeric descriptions are published with all their details in [48]. The random numbers from the turbulence model allow us to simulate different realizations of the fiber behavior.

For the simulation of the fiber dynamics, the software tool “FIDYST” was used, which was developed at the Fraunhofer ITWM, and the commercial CFD software “*Ansys Fluent*” was used for the fluid flow simulation. To restrict the computational effort while still considering all relevant turbulent effects, a shear stress transport (SST) k-ω turbulence model was used. All walls were seen as no-slip walls while neglecting heat transfer. For the process air inlet, the pressure and temperature were prescribed, and an ambient pressure of 30 °C was assumed for the domain boundaries.

## 3. Results and Discussion

### 3.1. Material Characterization and Suitability Characterization for the Nanoval Process

The criteria to define a polymer as processible with the Nanoval process were defined as follows:Polymer melt exits the die continuously and is taken up by the air stream without forming adhesions at/around the orifices, which retain at the orifice or “fall” discontinuously on the deposition.
○Lower limit of process window; criteria: all orifices are free of adhesion.
Polymer melt stream hits the conveyor too hot/too low in viscosity, which means no fibers are deposited due to the merging of the melt on the belt, respectively.
○Upper limit of the process window; criteria: fibers are deposited, and a coherent web can be wound up


Therefore, the process temperature was used as the first criteria to find a general production window; polymer throughput and air amount, respectively, and the ratio of both were set as the second criteria to generate a sharper insight into possible process conditions and/or their limitations. As the default setting, a polymer throughput of 16.4 kg·h^−1^ (3.80 g·ho^−1^·min^−1^) and an air amount of 214 Nm^3^·h^−1^ were set. In the case of adhesion of melt on the orifice, the air amount was increased up to max. 440 Nm^3^·h^−1^ in a first step, and the polymer throughput was lowered down to 1 kg·h^−1^ (0.23 g·ho^−1^·min^−1^) in the following (if necessary). In addition to the qualitative evaluation of the process, the polymers were characterized in a shear-rheological experiment. Therefore, the onset of process limits was set in relation to the complex shear viscosity, measured in temperature-sweep mode. This is shown for the three polypropylene types in Figure 3.

It was found that the onset of adhesion (lowest possible process temperature) of melt at the spinneret orifice correlated with a certain viscosity level of the respective polymer as well as the maximal process temperature (set by the formation of a melt film on the conveyor belt). Here, the viscosity obtained by the characterization at 10 rad·s^−1^ angular frequency can be seen as the zero-shear viscosity (see plot of complex viscosity vs. shear rate for the different polypropylenes for different temperatures in the Appendix A). The viscosity being present in the process is not available as the occurring shear rates are not accessible in plate–plate rheological experiments, so the zero-shear viscosity can be used as an easily accessible and traceable reference value. Dependent on the differences in the molar mass distribution of the three polypropylene types (MFI of 35 g·10min^−1^ (HH450FB), 400 g·10min^−1^ (HL504FB), and 1200 g·10min^−1^ (HL712FB)), die adhesions occurred at different process temperatures, each with a level of the complex viscosity of 15–18 Pa·s, with a shift to higher temperatures (HL712FB: 200 °C, HL504FB: 250 °C, HH450FB: 345 °C) the higher the molar mass. The highest possible process temperature behaves exactly the other way around, which corresponds to a level of complex viscosity around 2–3 Pa·s and occurs earlier for the two lower-viscous meltblown types (305–315 °C) than for the high-viscous spunbond type *HH450FB*. It must be noted that this limit is also influenced by the setup, especially by the construction height of the spinning beam and thus the resulting maximal adjustable DCD-ratio (DCD_min_ = 750 mm).

This viscosity range for the processability was found to be independent of the polymer type and is also transferable to estimate the process-temperature window of the Nanoval process for each of the other polymer types (PET, PBT, PA6, PPS, and PEEK). One restriction at this point was to set maximum temperatures for the processing of the polyesters and polyamides due to their characteristic temperatures for thermal degradation. So, a maximal processing temperature of 290 °C for PBT, 330 °C for PET, and PA6 were set, which are quite tough compared to standard processing conditions and already above the respective onset temperatures for thermal degradation. The temperature sweeps of all tested polymers with inclusion of the found viscosity limits for processing are shown in Figure 4.

The found temperature viscosity criteria could be used successfully to point out the suitable range of the two different molecular grades of PBT (Pocan B600 and Celanex 2008) and PET (RT5140 and Advanite 64001), respectively. As shown in the viscosity-temperature plot, the curves of higher viscous types are slightly shifted toward higher viscosity due to the higher molar mass (indicated by the IV or the MFI). Pocan B600 and RT514*0*, both theoretically reach the viscosity window for processing above the critical temperature for degradation (PBT: 290 °C/PET: 330 °C). The same unsuitable processing area exists for the PA6-type *B24NO3*, which reaches the viscosity window at 335 °C and is thus both too critical for extrusion and also shows indications of interaction between degradation and crosslinking (molecular build-up; see G′/G″ vs. temperature plot in the Appendix A). Nevertheless, it was possible to lay down nonwovens on the conveyor belt experimentally, but with a significant amount of shots. These shots could not be compensated at all by varying other process parameters (e.g., higher air temperature or air flow, lower DCD, or lower polymer throughput).

Summarizing, the following polymers were characterized and tested in the process but were not processable:Pocan B600 (PBT);RT5140 (PET);B24N 03 (BASF).

As the viscosity of the PEEK-type Victrex 90G lies above 100 Pa·s, still at 450 °C, which is also the temperature limit. Thus, PEEK was discarded from processing, so the following polymers were used to finally evaluate the Nanoval process. In detail: Borealis HL712FB, HL504FB, and HH450FB (PP), Celanese 2008 (PBT), Advanite 64001 (PET), and Fortron 0203 (PPS).

Due to the harsh process conditions the polymers are exposed to in the Nanoval process, the degradation during the nonwoven deposition process was characterized using the PP materials as an example and is given in Appendix B.

### 3.2. Experimental Characterization of the Nanoval Process

After revealing the material perspective, the Nanoval process itself was characterized in the context of further existing technologies to produce polymer nonwovens. Taking up the results from Section 3.1, the processable polymer types were compared regarding their productivity in terms of maximal polymer throughputs and, in addition, the achievable range of fiber diameters within the process window applied to the amount of air-to-polymer throughput (per filament). Figure 5 gives plots of the mean fiber diameter as a function of different process parameters in comparison of different polymer types and process conditions.

Basically, the majority of the process conditions lead to mean fiber diameters in the range of meltblown nonwovens. Moreover, the results meet the expectations concerning the increase in fiber diameter with increasing polymer throughput (Figure 5a) and finer fiber diameters with increased amounts of process air (Figure 5b) [2]. However, the high fiber diameters achieved with a polymer (type) of higher viscosity (Figure 5b) are of high interest as they exceed the typical range of meltblown noticeably. Further, the comparably stronger increase in fiber diameter with increasing throughput of the polyesters PBT and PET, to be derived from their more complex molecule structure, is of note and has to be counterbalanced by higher air amounts (higher ratio of air to polymer throughput per nozzle) to produce the same fiber fineness at higher productivity.

Additionally, an increase in the melt temperature (Figure 5c) and thus reduced viscosity lowers the fiber diameter, but with a higher risk for thermal polymer degradation and lower nonwoven mechanics, respectively. Furthermore, the possible temperature variability for the non-polyolifinic materials is narrow.

As shown in Figure 5d, the DCD ratio has a minor impact on fiber formation. At DCDs above 350 mm, which is the nearest possible setting with use of the air channel between die and collector, the fibers are not stretched to lower diameters anymore. In addition, the positioning of the air channel with respect to the distance from the die shows no significant impact (see plot of mean fiber diameter vs. distance of die to air channel (DDAC) for two different DCDs in Appendix A).

Using the variability of melt temperature, air amount, molar mass (molecular grade), and polymer throughput, the flexibility in processing PP and its influence on fiber diameter can be monitored. Thus, in Figure 6, the plot of the mean fiber diameter towards increased throughput is shown for the “Border areas” of the process windows as well as for further selected process settings.

For the same polymer at the same processing conditions, all curves show a linear relationship between throughput and diameter. The curves of the high-viscous PP HH450FB with higher air amounts are almost identical to those of the low-viscous HL712FB at a lower process temperature. This points out that the Nanoval process can be used to make different polymer grades processable, resulting in the same diameter range on top of it. This is different to the meltblown process as well as the spunbond process, which can only “handle” one polymer grade (range of viscosity) and result in far different diameter ranges.

To point out the flexibility of the Nanoval process, the maximal and minimal fiber diameters that could be achieved on average are shown in Table 3 and Table 4.

Focusing on the polypropylenes, the range of 0.80 μm (REM-image see Figure 7a) up to 39 μm (REM-image see Figure 7b) only by process setup adaptations without changing the spinning equipment is unique and combines the possibilities of meltblown and spunbond technologies:Highest fiber fineness was achieved with the low-viscosity type PP, maximal process temperature and air throughput, and minimal polymer throughput;Highest fiber coarseness was achieved with medium-viscosity type PP, minimal process temperature and air throughput, and maximal polymer throughput.

As stated before, the DCD and position of the air channel are of minor relevance, at least for the resulting fiber diameter. The deposition homogeneity will be considered separately in Section 3.4. Related to identical process conditions, the influence of the molar mass on the median and mean fiber diameter, the standard deviation, and the die pressure are given in the Appendix A (see Appendix A). Further, the high-viscosity PP (HH450FB) was not suitable to achieve high fiber diameters due to its limited variability of parameters and its rather extreme process conditions. However, the maximally achieved mean fiber diameter of 14.46 μm with the low-viscous type HL712FB does not represent the limit of the technology, as does the minimally achieved mean value of 2.17 μm for HL504FB.

In comparison to PP, the minimally achieved fiber diameters of the non-polyolefins are feasible in the range between 1 μm and 2 μm, and thus, in a comparable range as well as the corresponding die pressure. For the polyesters, the high pressure drop at comparable low throughputs is significant and corresponds to the maximally achieved fiber diameters, which do not arise to the same extent in other common melt spinning technologies.

To classify the Nanoval process economically, one of the biggest impacts is its mass output. Therefore, the maximal productivity is shown in Table 5 for all processed polymers in addition to the process window of the applicable melt temperature. Limited as usual by the defined maximal permissible die pressure, in our experimental work at 100 bar.

A maximal productivity of 110 kg·h^−1^·m^−1^ or corresponding 12 g·ho^−1^·min^−1^ could be reached with the meltblow PP-grade HL712FB, which was limited by the possible output rate of the used extrusion system while the die pressure was just around 55 bar. This pressure level is still far above the maximal pressure resistance of conventional meltblow dies, which ranges around 20–40 bar, and is caused by the design of fine capillary holes close together in a row at the tip of the die. This is an advantage of the Nanoval system against Exxon-type dies, beneath the higher flexibility in the selection of usable polymers. 

Additionally, the specific energy consumption for the Nanoval process at this throughput range can be pointed out. A total electric power consumption (extrusion system + air supply system + Nanoval spinning beam) of 48 kW was recorded at the used process setup. With the largest polymer throughputs of 55 kg·h^−1^ (for a 0.5 m line width), a value of 0.91 kWh·kg^−1^ results. This value is superior compared to the laboratory meltblow system of the same width, and also 70 kW of energy consumption for an output of 33 kg·h^−1^ of PP HH450FB at 345 °C melt temperature results in a specific value of 2.2 kWh·kg^−1^, when furnishing with meltblow lines.

The maximal applicable productivity of the type HL504FB can be assumed to be much higher than represented, due to the fact that the maximal throughput was set rather by exceeding the pressure limit than by the extrusion system in this case. As mentioned before, the high-viscous type and the non-polyolifinic polymers show a reduced process productivity due to the significantly higher pressure level, the latter related to the more complex molecular architecture, and the lower temperature flexibility concerning the onset of degradation and crosslinking. This also has an impact on the maximal and minimal fiber diameters shown in Table 3 and Table 4, which could be achieved on average.

Summarizing the result of the trials, the ratio of air to polymer throughput (per filament) can be identified as a major effect parameter for the fiber diameter, which is, as expected, supplied by the melt temperature, especially to reach high fiber diameter values. A minimal fiber diameter average is achieved at a ratio of around 4 Nm^3.^kg^−1^, and a maximal fiber diameter average at a ratio below 0.20 Nm^3.^kg^−1^. Both values were not achievable for all of the polymer types, especially when limited by the used compressor system on the one hand and by exceeding the pressure limit (reaching the maximal productivity) on the other hand.

Along with the adjustment of the fiber diameter, the deposition layout of the nonwoven fabrics also changes with the fiber fineness/coarseness. As depicted in Figure 8, the deposition homogeneity also aligns with spunbond-like fabric deposition (Figure 8a), going along with the more spunbond-like process conditions (process temperature, throughput, low/no initial process air application). The most significant difference is the absence of a quench duct to initiate the spunbond-typical stretching of the fibers to apply the characteristic high mechanical strength. However, this could be additionally integrated into the Nanoval process by replacing the “passive” air channel with an “active “quench duct”, applying a necessary air speed of ~ 200 m·s^−1^ and by using a single Nanoval capillary row, which additionally forms only a minimal set-back in between the capillary tip and the opening of the spinning beam (leaving a minimal air flow cross-section in the Laval-die).

The more the process conditions are turned towards meltblown process settings or even further (higher process temperature, lower throughput, high air stream), the softer the haptic of the fabric, and the more homogeneous the deposition results (Figure 8c,d).

### 3.3. Simulation Results of the Nanoval Process

The air flow of the Nanoval process is simulated by dividing the area into subdomains of different accuracy and making use of periodicity. The air supplies for the nozzles are spatially separated, so it makes sense to compute these air flows separately. There are two different kinds of geometries for the air supplies. One geometry for the two rows in the center, and one for the rows on the outside. This means that it is sufficient to compute only two representative simulations instead of one simulation for each nozzle. For both geometries, the air flow enters from the side and is then led in a downward spiral towards the spinning direction. Figure 9 shows the geometry of the center rows with streamlines starting from the air inlet. The polymer flows in the center of the pink capillary and enters the air stream at its tip. The end of the transparent blue volume is the exit of the spinning beam. The inlet air pressure for the visualized result is 0.75 bar (corresponding to 220 Nm^3.^h^−1^ throughput) and the temperature is 294 °C. One can see how the air is accelerated during the process due to the expansion at the exit of the nozzle. The figure also shows how much swirl is applied to the downward-facing air stream. The air stream conditions at the tip of the pink capillary are of special interest since they are responsible for the first forces acting on the fiber.

The downward spiral is one of the main differences compared to meltblown processes and was therefore a focus of the CFD analysis. The comparison did show a significantly higher vorticity for the Nanoval process, with less applied air throughput for the first few millimeters after the polymer exits the nozzle. A visual representation of the comparison is shown in Figure 10, where the left graphic (Figure 10a) is the Nanoval process with the same boundary conditions as in the previous figure (Figure 9), and on the right side (Figure 10b) is a meltblown process with an air pressure of 1 bar (corresponding to 325 Nm^3.^h^−1^ throughput) and a temperature of 412 °C. In both graphics, a plane through the center of one capillary is colored by the vorticity.

As soon as the air streams exit the nozzle, a strong interaction between the streams occurs since they all draw secondary air from the same reservoir. Therefore, the streams can no longer be seen as independent but must be simulated within one setup. For this simulation, the periodicity is exploited, and only three columns of nozzles are computed with periodic boundary conditions in the cross direction. This does not capture the boundary effects in the cross direction but is an appropriate model for all columns in the center region. The air flow conditions from the detailed simulation of the nozzle serve as inlet boundary conditions for the main periodic simulation. The interface is located at the cross section where the polymer enters the airstream and includes the local velocity vector, temperature, turbulent kinetic energy, and turbulent dissipative rate. The main simulation includes the whole fiber-forming domain with the deposition belt and the suction box underneath. For the Nanoval process, it is possible to change the direction of rotation for certain rows. The assumption was that rows with opposite directions of rotation would amplify their vorticity. To verify this assumption, three different scenarios have been simulated, where the direction of rotation is the same for all rows, is alternating from row to row, and the center rows rotate opposite to the outer rows. Figure 11 shows the vorticity in a horizontal plane 3 cm underneath the spinning beam for all three scenarios. It is notable that the magnitude of the vorticity is less in the outer rows and that the affected area is more feathered out. However, this is the case for all scenarios, which means that the vorticity cannot be controlled by the direction of rotation.

The results of these fluid flow simulations are the basis for the simulation of fiber dynamics. As we have seen, the Nanoval process has higher turbulence in the air flow at the start of the fiber-forming region than a traditional meltblown process. Therefore, the stationary part in the fiber simulation was reduced to 3 mm, which was also indicated by observations and high-speed recordings in the experiments/real process. Figure 12 shows the result of one fiber simulation for a certain timestep, where the uniaxially viscous part is colored in blue and the unsteady viscoelastic part is colored in red. The turbulence has a huge effect on the deflection of the fiber, which is shown on the left side. Likewise on the right side, the diameter along the height is visualized. One can see that the diameter is decreasing right after the polymer exits the nozzle, but the decrease becomes strongly accelerated as soon as the turbulence can affect the fiber.

In general, the simulation did overestimate the resulting end diameter, but it can predict tendencies and fiber diameter distributions very well. In Figure 13, the simulation results are compared to measurements for two different scenarios. The measurements are colored green, and the other colors represent the four different spinning positions. The difference between the two scenarios is an increase in air throughput from 220 to 440 Nm³·h^−1^. One can see that the higher air throughput leads to thinner fiber diameters but also to a narrower diameter distribution for the measurement as well as for the simulation. The offset of the absolute diameter between simulation and measurement might be caused by the fact that surface tension was not taken into account in the simulation model. Ongoing research activities where surface tension is introduced into the simulation model show promising first results.

### 3.4. Effect of Active Air Guidance on the Deposition Homogeneity

Beneath the achievable fiber diameters, the deposition homogeneity is of high relevance as a main criterion for a nonwoven fabric and is highly influenced by the control of the process air and its turbulent behavior, as shown before in the simulations of the process. Figure 14a shows the area base weight, measured along the cross direction in two different sampling sizes, for a fabric produced with a 45 g·m^−2^ target value (= throughput/(winding speed * deposition width) in a “free” spinning room without the use of any additional setup.

Clearly recognizable is the fact that the finer sampling size of 0.785 cm^2^ rondes reveals a higher scattering than the testing of 100 cm^2^ samples according to the standard. This can also be seen in the coefficient of variation, which is 44 for the standard test and 79 for the more detailed method. The air permeability measured at three positions goes along with the base weight indirectly and proportionally. The finer sampling shows that the air guidance is insufficient as a majority of fibers are deposited in the middle, with a difference of almost 40 g·m^2^ at maximum to the edges.

The addition of hole sheet linings (Figure 14b) in the space between the spinning beam and deposition belt and at its sides to avoid or rather reduce air back- and side-flow qualitatively reduces “fiber flow”, with a slight effect on the values of area weight and air permeability. However, the coefficient of variation could be reduced by this measure.

Of greater effect is the addition of an air channel in the spinning space, which should guarantee the reduction in side and backflow of primary process air and additional help to control the inflow of secondary air into the system. As shown in Figure 14c, the variation coefficient decreases to 35 for the fine sampling size method, and the single data points reveal that the inhomogeneity towards the edges is optimized. However, the base weight is slightly below the target value due to the intervention in the fiber flow and the compensation of the deficiency at the fabric edges. However, this issue can be easily compensated by adjusting the winding speed [2].

### 3.5. Simulation of the Secondary Air Flow

The design of the air channel has been supported by CFD simulations. The behavior of the air flow in the fiber-forming zone is highly transient, and an unsteady simulation setup was used. For the presentation of the results, only one snapshot of the simulation is used. Figure 15 shows the impact of the air channel on the streamlines, starting from the outer boundaries and the nozzles. The figure shows at the top the four spinning rows with the primary air and the secondary air, which is drawn from the sides. With the help of the air channel, it was possible to control the amount of secondary air in the system, but it also caused huge transient swirls that made the process unstable. This led to the development of an air channel with perforated walls, which prevent the formation of these swirls.

Another challenge of the Nanoval process is to control the behavior at the fabric edges and the fiber flight. To be able to analyze these topics with the help of the simulations, the model had to be expanded to the full depth of the plant, including all spinning positions. Figure 16 shows the streamlines starting from the nozzles for a scenario where lots of fiber flight was detected during the experiment. The left graphic shows that not all streamlines from the nozzle exit through the suction box. Some of them leave the domain through the boundaries on the sides, causing fibers to fly free in the room. The simulation was used to compute the necessary decrease in pressure at the suction box to overcome this effect. The result is shown on the right-hand side of the graphic, where the suction pressure was reduced so that about four times more mass exits through the suction box.

## 4. Conclusions

Experimental results show that the Nanoval technology is suitable to manufacture nonwovens from polymer grades with a range of complex viscosities of 2–18 Pa*sand an MFI of 40 g·10min^−1^ up to 1200 g·10min^−1^ regardless of the polymer type (polyolefins, polyesters, and high temperature polymers). For polypropylene, the resulting fiber diameter could be shown in the range of below 1 to 40 µm without changing the spin-beam equipment, whereas the nonwoven webs are on the one hand comparable to meltblown webs and on the other hand comparable to spunbond webs. This wide range of achievable fiber diameter is applicable to other polymers likewise; for example, polyester shows a fiber diameter range of 1.5 to 15 µm without changing the equipment or PET grade.

Moreover, the different design of the *Nanoval system* provides, on the one hand, much higher productivity with polymer throughput > 110 kg·h^−1^·m^−1^ or corresponding 12 g·ho^−1^·min^−1^ (PP-grade HL712F*B*) compared to Exxon-type dies (meltblown) due to higher pressure resistance, and, on the other hand, a distinct higher flexibility in the selection of usable polymers in comparison to both meltspun technologies (meltblown and spunbond). 

A further benefit of the Nanoval process is its low specific energy consumption (kWh·kg^−1^), resulting in savings of around 60% compared to meltblown lines at the same processing conditions.

As with the meltblown procedure, the Nanoval process is very complex, with many interacting process parameters. Therefore, an optimum combination of these influencing variables is necessary for a trouble-free production. However, the Nanoval technology has the potential to operate according to both processes, especially on a smaller scale than industrial production lines, for (quantitative) niche applications in every nonwoven sector. In addition to the general application potential of the production of high-temperature resistant nonwoven media in the fields of filtration and separation, fuel cells, battery separators, novel underlay membranes, technical textiles, and many more, a new product field arises for the application of Nanoval technology with regard to the resulting fiber fineness, which can close the gap between meltblown and spunbond technology or combine them in one production plant.

The wide range of parameter studies supported by the simulation results leads to a deeper understanding of the influence and interaction of different process parameters. Although the simulated results differ from the measured ones in absolute terms, the relative tendencies have been correctly predicted by the simulations of the Nanoval process. This allows the finding of optimal parameter settings to obtain the best final product properties. Ongoing research shows that the inclusion of surface tension in the fiber model tends to deliver thinner diameters and better consistency with experimental data. The extension of the model is planned for future work. In addition to the fiber diameter predictions, valuable insights were gained from the computational fluid dynamics simulation. It highlighted the advantages and disadvantages of a closed air channel and resulted in the final perforated air channel. Furthermore, the optimal flow through the air suction was simulated to generate a stable process.

## Figures and Tables

**Figure 1 materials-16-02932-f001:**
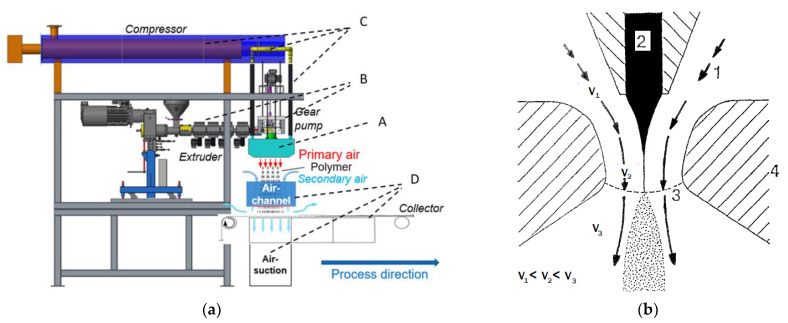
Scheme of the Nanoval process; (**a**) main components of the extrusion and air system of the Nanoval pilot plant at DITF Denkendorf; A: Nanoval spinning beam, B: Extrusion system, C: process air supply, D: air channeling/suction and nonwoven deposition; (**b**) Principle of the polymer and air flow for one spinning hole; 1: (hot) process air stream/channel; 2: polymer flow; 3: spinnhole; 4: die block.

**Figure 2 materials-16-02932-f002:**
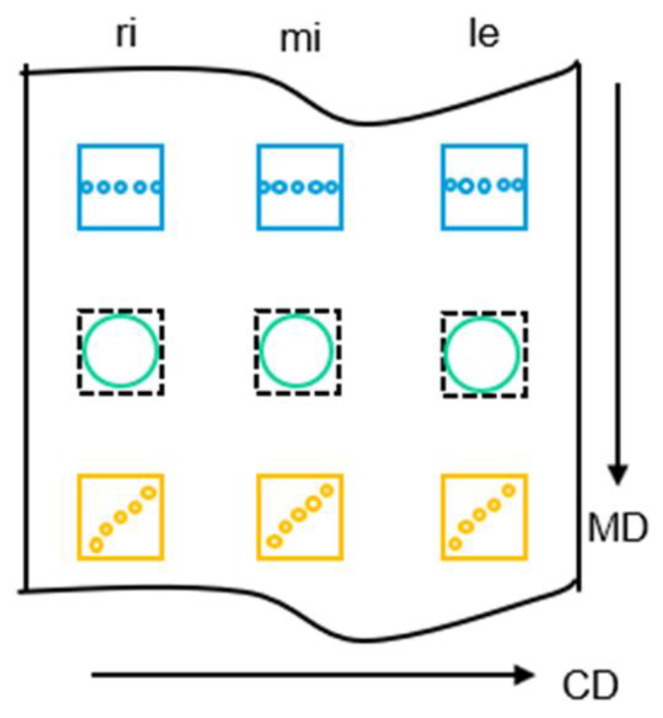
Scheme of the sampling on the produced nonwovens for analysis of nonwoven properties and homogeneity (distribution); squares: 10 × 10 cm for area base weight measurements; blue dots: sampling of area base weight in finer resolution (0.782 cm^2^); green circles: air permeability (20 cm^2^); yellow dots: thickness measurements (1 cm^2^); abbreviations: machine direction (MD), cross direction (CD), left (le), middle (mi), and right (re).

**Figure 3 materials-16-02932-f003:**
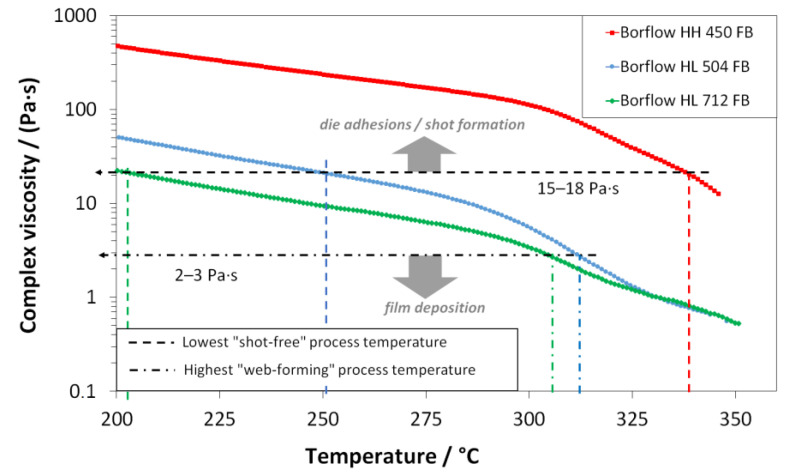
Shear rheological temperature sweeps (ω = 10 rad·s^−1^, ε = 10%, $\dot{T}$ = 0.5 K·min^−1^) of three polypropylene types (red: HH450FB; blue: HL504FB; black: HL712FB) and onset temperatures of process limitations (dotted line: min. process temperature; dot-line-dot: max. process temperature); the related plots of the storage- and loss modulus are given in the Appendix A (see Appendix A).

**Figure 4 materials-16-02932-f004:**
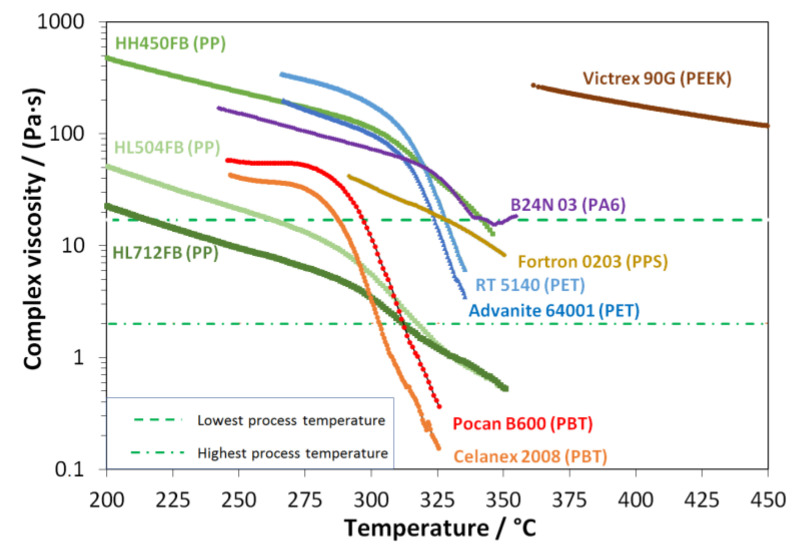
Process temperature windows of different polymer types for the Nanoval process, defined by the shear rheological temperature sweeps (ω = 10 rad·s^−1^, ε = 10%, $\dot{T}$ = 0.5 K·min^−1^) and observed onset temperatures of process limitations (dotted line: min. process temperature; dot-line-dot: max. process temperature); the related plots of the storage- and loss modulus are given in the Appendix A (see Appendix A).

**Figure 5 materials-16-02932-f005:**
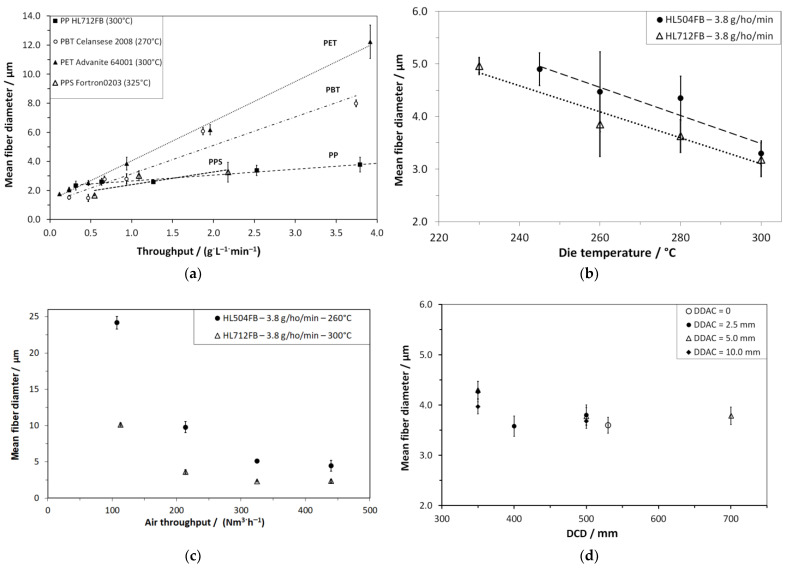
Mean fiber diameters as a function of different process parameters: (**a**) vs. polymer throughput (PP HL712FB, PBT Celanese 2008, PET Advanite 64001, and PPS Fortron 0203; $\dot{V}$_air_ = 220 Nm^3.^h^−1^); (**b**) vs. air throughput (PP HL504FB and HL712FB, $\dot{V}$ = 3.8 g·ho^−1^·min^−1^); (**c**) vs. die temperature (PP HL712FB and PP HL504FB), $\dot{V}$ = 3.8 g·ho^−1^·min^−1^, $\dot{V}$_air_ = 220 Nm3·h^−1^) (**d**) vs. DCD and distance die-air channel (PP *HL712FB*, T_melt_ = 300 °C, $\dot{V}$ = 3.8 g·ho^−1^·min^−1^, $\dot{V}$ _air_ = 220 Nm^3.^h^−1^); the lines represent linear fits.

**Figure 6 materials-16-02932-f006:**
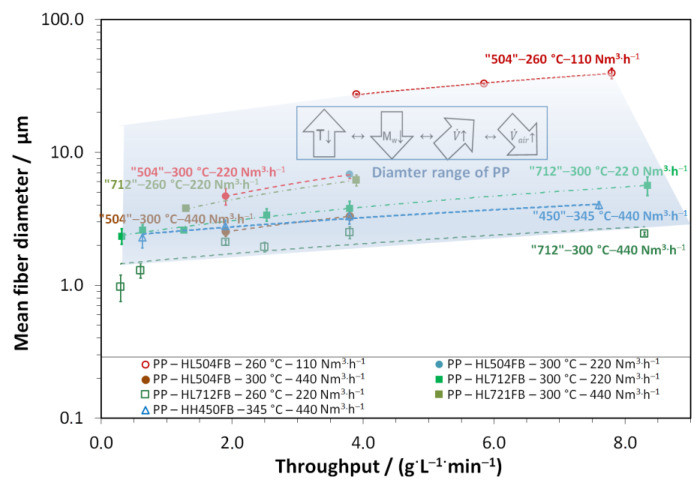
Process map of polypropylene for achievable fiber diameters as a function of the polymer throughput under variation of the PP-type, polymer, and air throughput (DCD, DDAC, and collector are kept constant); the lines represent linear fits.

**Figure 7 materials-16-02932-f007:**
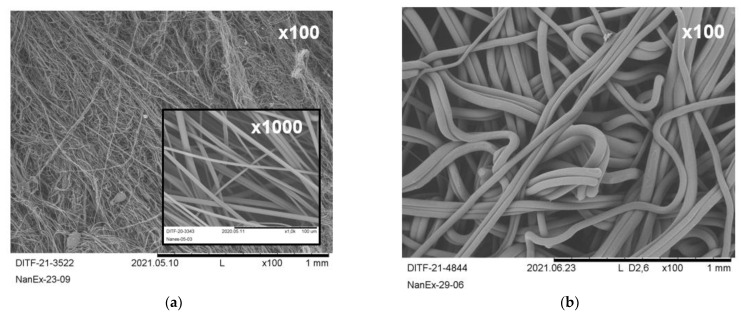
REM-images (×100 and ×1000) of PP fabrics with: (**a**) minimal median fiber diameter (0.8 μm); (**b**) maximal median fiber diameter (39.3 μm).

**Figure 8 materials-16-02932-f008:**
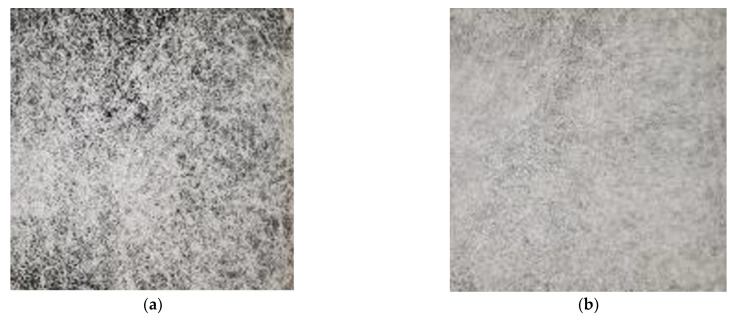
Fabrics of PP with decreasing average fiber diameter: (**a**) MD = 25 μm; (**b**) MD = 10 μm; (**c**) MD = 5 μm; (**d**) MD = 3 μm.

**Figure 9 materials-16-02932-f009:**
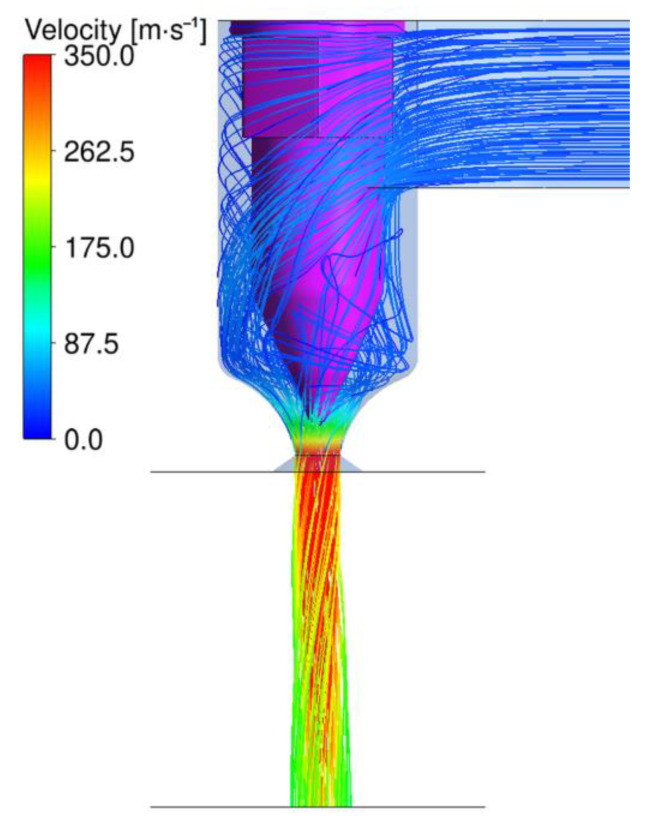
Simulation of air flow: streamlines for the center nozzles.

**Figure 10 materials-16-02932-f010:**
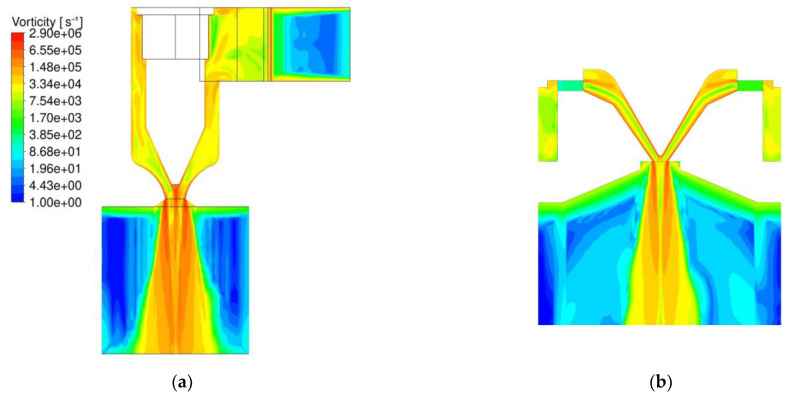
Comparison of the vorticity: (**a**) Nanoval process; (**b**) meltblown process.

**Figure 11 materials-16-02932-f011:**
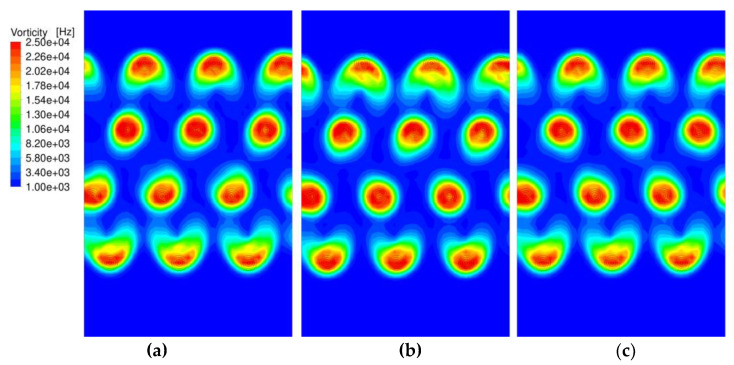
Simulation of the vorticity for different directions of rotation 3 cm underneath the spinning beam: (**a**) same direction for all nozzles; (**b**) alternating directions, (**c**) center rows rotate opposite to outer rows.

**Figure 12 materials-16-02932-f012:**
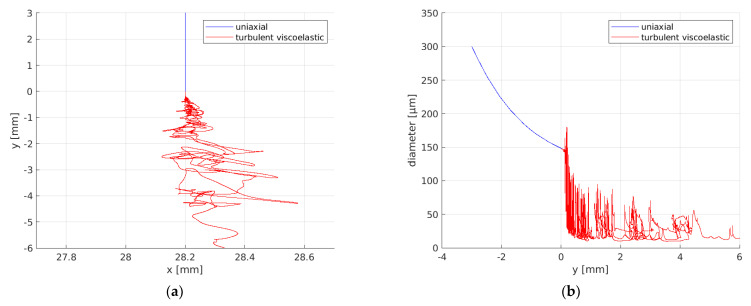
Uniaxial viscous (blue) and unsteady viscoelastic (red): (**a**) fiber deflection and (**b**) fiber diameter; simulated for a certain timestep.

**Figure 13 materials-16-02932-f013:**
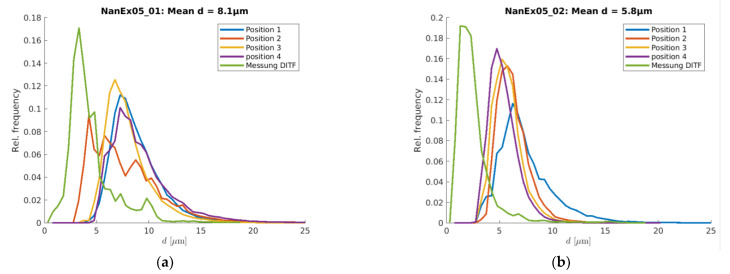
Comparison of diameter distribution of measurement and simulation for two scenarios, based on experiments with PPHL712FB (T_melt_ = 300 °C, $\dot{V}$ = 3.8 g·ho·min^−1^): (**a**) ${\dot{V}}_{air}$ = 220 Nm^3.^h^−1^; (**b**) ${\dot{V}}_{air}$ = 440 Nm^3.^h^−1^.

**Figure 14 materials-16-02932-f014:**
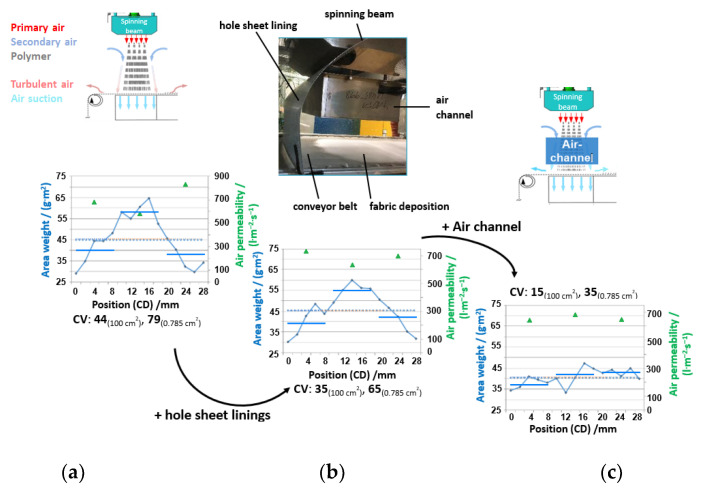
Fabric homogeneity along CD, characterized by area weight (sampling size 0.785 cm^2^ (14 measurement points: blue dots) and 100 cm^2^ (3 measurement points: blue lines)) and air permeability (three positions: green triangles) for spinning PP HL712FB at constant spinning conditions: (**a**) without add-on; (**b**) with addition of a hole shine lining to the space between spinning beam and deposition belt; (**c**) for addition of the air channel (200 mm length, distance 50 mm) between spinning beam and deposition belt.

**Figure 15 materials-16-02932-f015:**
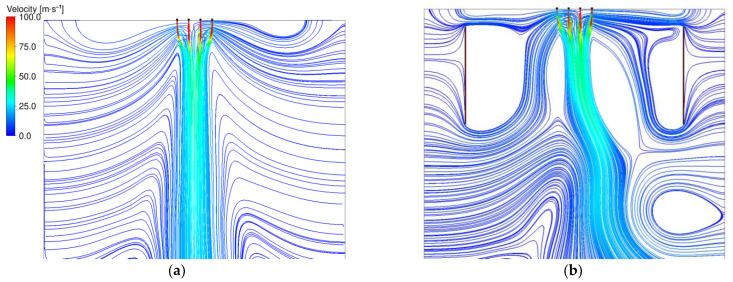
Simulation of the streamlines of process air, starting from the outer boundaries (secondary air) and the nozzles (primary air): (**a**) without air channel; (**b**) with closed air channel.

**Figure 16 materials-16-02932-f016:**
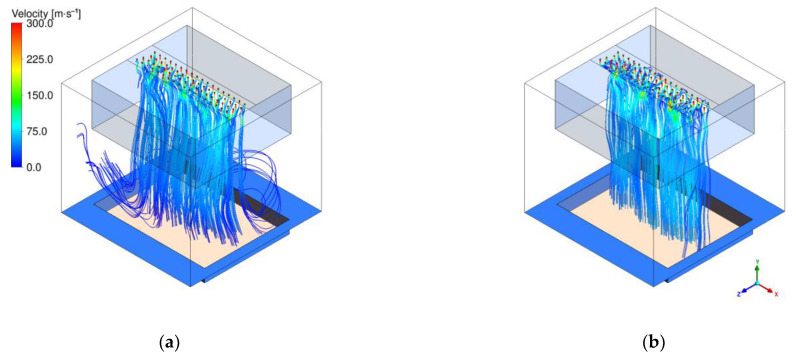
Simulation of process air streamlines, starting from the nozzles: (**a**) for a scenario with fiber flight; (**b**) the same scenario with optimized suction pressure.

**Table 1 materials-16-02932-t001:** Process specific material properties, process set-up specifications, and typical properties of meltblown and spunbond.

Property	Meltblown ^1^	Spunbond
Material:		
Polymer MFI range	>30–1500 g·10 min^−1^ [2,5]	12–70 g·10min^−1^ [2]
(Standard:)	>150 g·10 min^−1^ [5]	20 ^2^/35 ^3^ g·10min^−1^ [5]
Polymer dispersity:	Narrow [5]	Narrow [1,5]
Process setup:		
Capillary size:	0.2–0.5 mm ^4^	0.3–0.8 mm ^4^
Number of capillaries:	20–50 hpi ^5^ [2]	50–125 hpi [19]
	1000–2000 holes·m^−1^	1000–>6000 holes·m^−1^[6]
Capillary arrangement	Nozzles in single row	Nozzle matrix
		(rows × lines)
Web properties:		
Base weight:	1–400 g·m^−2^ [2]	10–800 g·m^−2^ [1,5]
Possible fiber diameters:	0.5–30 μm [5]	1–50 μm [1,5]
Typical fiber diameters:	2–7 μm [2,5,17]	15–35 μm [1,5]
Mechanical properties:	Low mechanical stability,	High tensile strength,
	Low abrasion resistance [2,5]	High abrasion resistance [1,3,5]
Further properties:	Filtering effect [2,5]	-

^1^ Exxon. ^2^ Europe. ^3^ USA. ^4^ Std ~0.4 μm [2,5]. ^5^ holes per inch.

**Table 2 materials-16-02932-t002:** Process-specific material properties, process set-up specifications and typical properties of meltblown and spunbond.

Polymer	Type	Supplier	Grade	T_m_ /°C] (ISO 11357-1-3/ISO 3146))	Density /(g·cm^−3^) (ISO 1183-1)	MFI /(g·10min^−1^) (ISO 1133-1)	(η) /(dL·g^−1^) (K070)	Other	Data Sheet
PP	Borflow^TM^ HL712FB	Boralis ^1^	fiber-type for meltblown applications and micro denier fibers at high spinning speeds	158	0.90	1200 (230 °C, 2.16kg)	-	-	[35]
PP	Borflow^TM^ HL504FB	Boralis ^1^	fiber-type for meltblown applications and micro denier fibers at high spinning speeds	161	0.90	450 (230 °C, 2.16kg)	-	-	[36]
PP	Borflow^TM^ HH450FB	Boralis ^1^	fiber-type grade for spunbonded nonwovens	161	0.90	35 (230 °C, 2.16kg)	-	-	[37]
PBT	B600 (TP010-002)	Lanxess ^2^	for extrusion and injection molding with improved flowability	225	1.310	171 ^3^	-	-	[38]
PBT	Celanex 2008	Celanese ^4^	melt blown applications	225	1.310	280	-	-	[39]
PET	RT5140	Invista ^5^	-	-	-	-	0.65 ^6^	-	[40]
PET	Advanite 64001	Advansa ^7^	-	-	-	-	0.550	-	[41]
PA6	Ultramid B24N 03	BASF ^8^	fiber-type grade for high-speed spinning	220	1.12–1.15	-	.	Relative viscosity (ISO394) 2.43.	[42]
PPS	Fortron 0203HS	Ticona ^9^	very easy flowing, heat-stabilized melt blown type	280	1.35	-		-	[43]
PEEK	90G	Victrex ^10^	-	343	1.30	-		Viscosity: 90 Pa·s at 400 °C (ISO 11443)	[44]

^1^ Borealis Polymere GmbH (Burghausen, Germany). ^2^ LANXESS Deutschland GmbH (Köln, Germany),^3^ calculated using MVR and density [38]. ^4^ Celanese Services Germany GmbH (Sulzbach, Germany). ^5^ INVISTA Resigns and Fibers (Gersthofen, Germany). ^6^ 1% solution in dichloroacetic acid.^7^ Advansa GmbH (Hamm, Germany). ^8^ BASF SE (Ludwigshafen, Germany). ^9^ Ticona GmbH (Sulzbach, Germany). ^10^ Victrex plc (Lancashire, United Kingdom).

**Table 3 materials-16-02932-t003:** Maximal accessible * fiber diameters of the processed polymer types and related process parameters.

Polymer	Fiber-Diameter		Melt-Temperature	Throughput	Throughput-	Die
	Mean	Stdev.			Ratio	Pressure
-	/μm	/μm	/°C	/(g·ho·min^−1^) /Nm^3^h^−1^)	/(Nm^3.^kg^−1^)	/bar
*PP HL712FB ***	*14.46*	*0.99*	*240*	*3.9/106*	*0.09*	*29*
PP HL504FB	39.33	3.57	255	7.8/102	0.04	57
PP HH450FB	4.28	0.43	350	7.6/465	0.20	88
PET Advanite 64001	14.39	5.63	300	3.9/220	0.18	96
PBT Celanex 2008	8.65	1.03	270	3.7/220	0.19	88
PPS Fortron 0203	4.29	0.68	325	2.2/220	0.37	79

* Limit set of current (used) system, not of the technology. ** Limit not reached as no higher throughput was used in trials for low air amounts.

**Table 4 materials-16-02932-t004:** Minimal accessed fiber diameters of the processed polymer types and related process parameters.

Polymer	Fiber-Diameter		Melt-Temperature	Throughput	Throughput-	Die
	Mean	Stdev.			Ratio	Pressure
-	/μm	/μm	/°C	/(g·ho·min^−1^) /Nm^3^h^−1^)	/(Nm^3.^kg^−1^)	/bar
PP HL712FB *	0.80	0.22	300	0.32/440	4.48	15.2
*PP HL504FB ***	*2.41*	*0.13*	*300*	*3.80/440*	*0.77*	*34.0*
PP HH450FB	1.90	0.22	355	0.60/440	2.24	14.4
PET Advanite 64001	1.51	0.18	300	0.20/290	3.97	19.1
PBT Celanex 2008	1.77	0.06	270	0.47/440	3.11	20.2
PPS Fortron 0203	1.91	0.09	325	0.54/220	1.47	25.0

* Limit set of current (used) system, not of the technology. ** Limit not reached as no lower throughput was used in trials.

**Table 5 materials-16-02932-t005:** This is a table. Tables should be placed in the main text near to the first time they are cited.

Polymer	Process-Window	Maximal Polymer Output		Die Pressure
	(Temp.)/°C	/(g·ho^−1^·min^−1^)	/(kg·h^−1^·m^−1^)	/bar
PP HL712FB	230–300	12.6	110	55
PP HL504FB ^1^	245–310	7.6	66	57
PP HH450FB	345 (+) ^2^	7.6	66	88
PET Advanite 64001	300	3.9	34	100
PBT Celanex 2008	270	3.7	32	83
PBS Fortron 0203	325+	2.2	16	79

^1^ Limit not reached as no higher throughput was used in trials for low air amount. ^2^ Higher temperature possible, but not applicable due to process safety issues (ignition temperature) [37].

## Data Availability

Data available on request due to privacy restrictions. The data presented in this study are available on request from the corresponding author. The data are not publicly available due to running project issues.

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
