# Peer review of "Nanoval Technology—An Intermediate Process between Meltblown and Spunbond"

_materials, 2023, doi:10.3390/ma16072932_

Round 1

Reviewer 1 Report

              The manuscript entitled “Flexible creation of fine and coarse fiber diameters in meltspun nonwovens using the Nanoval Laval-die process as an intermediate technology between meltblown and spunbond processes” by Höhnemann et al. investigates the feasibility of employing the Nanoval technology to manufacture nonwovens from different commodity and engineering thermoplastics. The authors evaluated the processing window for fabricating defect-free nonwoven webs that are similar to the melt-blow and spunbond counterparts. Specifically, using polypropylene a wide range fiber diameter was obtained without the need of significant process modifications allowing for the manufacturing of webs for different applications using the same experimental setup. Further, numerical simulations accounting for the complex process physics (flow, temperature, and viscoelasticity) were performed to validate the experimental trends and they were found to be in qualitative agreement with each other. Discrepancies between the model and experimental findings could potentially be addressed by incorporating surface tension and inertia effects that drive the web formation process. The work performed is original and is of significant interest to the broad readership of the journal. However, there are a few issues associated with the manuscript, as outlined below, that need to be addressed before it can be considered for publication in the Materials journal.

Comments to the Authors:

1.     There are a few relevant literature reports that are missing in the manuscript. A non-exhaustive list of such sources is provided below.

a.     https://doi.org/10.1038/s41428-020-00446-y

b.     https://doi.org/10.3390/ma13194298

2.     In order to completely alleviate concerns regarding material degradation, isothermal time sweeps at a constant frequency can be performed.

3.     Please improve the resolution of Figure 1.

4.     Consider consolidating the Introduction section of the manuscript.

5.     Section 2.5: Usually strain sweeps are performed at constant angular frequency; why did the authors perform it under constant shear rate? Edge fracture can be an issue in such cases.

6.     Why were the temperature ramps performed under constant gap mode? With increase in temperature, the material is likely to flow and contact between plates and the sample decreases. A better way to run temperature ramps in the rheometer is under the constant normal force mode to ensure contact throughout the testing interval.

7.     Please provide the details about the abbreviations used in Figure 3 in the figure captions.

8.     What was the accelerating voltage and operating mode in the SEM? Please provide more details regarding sample preparation post-SEM imaging.

9.     Please provide the storage and loss moduli data from the rheology testing and tie back the shape fidelity of the fibers to the observed experimental trends.

10.  Resolution of Figure 15 is unsatisfactory.

Author Response

We thank the reviewer for the detailed review and all the specific comments, which will help to eliminate different unclear or fuzzy passages, especially in the experimental methods. Please find following the detailed answers to all comments provided.

  1. There are a few relevant literature reports that are missing in the manuscript. A non-exhaustive list of such sources is provided below.
  2. https://doi.org/10.1038/s41428-020-00446-y
  3. https://doi.org/10.3390/ma13194298

Thank you for the recommendation. However, we consider our literature to be relevant to state the current stand of technique for a) meltblown and b) spunbond nonwoven processes. The stated additional literature addresses the melt spinning of yarns for textile applications (https://doi.org/10.1038/s41428-020-00446-y) and the melt-electrospinning process, which both are not directly related to the work we perform or the processes we consider. To additionally oppose the Nanoval process to yarn spinning or melt-electrospinning will introduce further which are needed to classify the process intermediate between meltblown and spunbond processes. In opposite, we think it will make the introduction more unspecified.

  1. 2.     In order to completely alleviate concerns regarding material degradation, isothermal time sweeps at a constant frequency can be performed.

Our aim is not to alleviate concerns on material degradation as we know that the process conditions in the Nanoval process are quite harsh considering the process temperatures (at least for PP). This we describe in the appendix in section A. Actually, we performed isothermal time sweeps for all materials at the relevant process temperature, which we were added to section A. (p.12, l.850ff)

  1. Please improve the resolution of Figure 1.

We replaced the figure by a version of higher resolution. New figure number is figure 1b and figure was shifted to page 5 due to comments of the other revisors.

  1. Consider consolidating the Introduction section of the manuscript.

We revised the Introductions section at various points also connected to comments of the other revisors. All changed passages are marked respectively. (p.2: l.53, l. 6, l.61f, l.67, l.74f, l.78f, l.83, p.3: l.99f, l.100, l.101, l.108, l. 120, p.3: l.125, l.126, l.128f, l.131, l.151-155, l.164, l.178-186)

  1. Section 2.5: Usually strain sweeps are performed at constant angular frequency; why did the authors perform it under constant shear rate? Edge fracture can be an issue in such cases.

Actually, we performed amplitude sweeps at 10 rad/s. We converted to shear rate afterwards to use a more common unit (s-1 / Hz) for the audience. The paragraphs (p.9 l.316 & p.8, l.319) were adjusted accordingly. We apologize for creation of confusion.

  1. Why were the temperature ramps performed under constant gap mode? With increase in temperature, the material is likely to flow and contact between plates and the sample decreases. A better way to run temperature ramps in the rheometer is under the constant normal force mode to ensure contact throughout the testing interval.

Actually, we performed the temperature sweeps under adjustment of the gap. (A normal decrease of gap was from 1.0 to around 0.79 mm) over the measurement time.   We added the missing explanation in the passage: “Temperature ramps ware performed under adjustment of the gap in order to maintain constant normal force over the measurement.” (p.9, l.3106f)

  1. Please provide the details about the abbreviations used in Figure 3 in the figure captions.

We added the explanation of the abbreviation used in the figure caption (p.9, l.330). New figure number is Figure 2 due to other changes in the manuscript.

  1. What was the accelerating voltage and operating mode in the SEM? Please provide more details regarding sample preparation post-SEM imaging.

The accelerating voltage of the SEM is 1 5kV and the device runs in “charge-up reduction mode”. We added this this to the method description (p.10, l.355). The details about sample preparation (voltage, current, time, vacuum) were added accordingly (p.10, l.351).

  1. Please provide the storage and loss moduli data from the rheology testing and tie back the shape fidelity of the fibers to the observed experimental trends.

In addition of the rheology testing shown in the manuscript, we added the according storage and loss moduli plots to the supporting information. However, we see at this point no connection to shape fidelity.

  1. Resolution of Figure 15 is unsatisfactory.

We replaced the figure by a version of higher resolution. New figure number is figure 14 due to other changes in the manuscript.

Reviewer 2 Report

Comments can be withdrawn from attached file. 

Author Response

We thank the reviewer for the suggestions, which will help to get the manuscript clearer for the readers understanding.

- General comments:

- Manuscript need to be revised to comply with the editing conditions of the journal. Sections numbering should be checked and ordered increasingly. Authors are advised to revise their main sections and to bring clear understanding on their reported data, both experimental and numerical simulated.

We revised the manuscript concerning the editing conditions of the Journal. We checked the section numbering and corrected the error. We apologize for the wrong numbering and confusion on this. The main sections have been revised as well according to the reviewer’s comments. We hope we meet now a higher understanding on our data. All changes are marked respectively in the manuscript.

- Title

- Title should be made more compact and attractive.

We changed the title to “Nanoval technology – an intermediate process technology between meltblown and spunbond” and hope the more compact version is more attractive. 

- Abstract

- Should be revised to emphasis better the subject approached, both qualitative and quantitative. Rows 14 to 18 should be revised from grounded.

We comprehensively revised the abstract. Especially rows 14 to 18 have been shortened also according to comments of the other revisory.

- Introduction

- Page 4, Figure 1 and rows 137 and 150 Figure should be omitted from this introductory part, mostly due to the fact that the manuscript was submitted under research article category and not book chapter or other scholar related contribution.

We omitted the figure and respective relating text passages from row 137 and 150 from the Introduction. However, we think it is very important for the reader to visualize the Laval die concept of Nanoval visually. So, we added the figure to figure 2 as figure 2b). New figure number is Figure 1a at p.6, l.199-211.

- Page 4, rows 164-167 and page 5, rows 168-182 Paper aims should be clearly formulated and must emphasis the main issues that will be approached through the subsequent sections.

We revised the respective paragraphs briefly. All changes are marked respectively. See p.6., l.178-186.

- Materials and methods

- Section 2.3 Entire section should be revised to bring clarity. We highly recommend gathering of information on material selection in tabular form.

We revised the entire section and transformed the material information in tabular form. See. P.7, l.264-p.8,l.270.

- Section 2.7 Section title should be adapted to the content. Section should be revised to bring clarity on the conditions deployed for simulation.

We have changed the section title to “Numerical description of the simulation model” which describes the content more precise and we have added the governing equations for the fiber dynamics to explain the background and limitations of the simulation model.

- Results and discussions

- Page 11, rows 417-441 Information provided are in relation with section 2.7. These should be revised to bring clear understanding.

The designated paragraph (p11., row 417-441, , now: p.12 l.452-476) is not in (direct) relation with section 2.7. Therefore, we did not consider changes.

- Page 16, section 3.3 Confusing and not in accordance with the content. This section should be considered distinctively, and prior the experimental results.

We changed the section title to “Simulation results of the Nanoval process”, which was the same as section 3.2 due to a copy-and paste error, which we didn’t notice. We apologize for this!

- Conclusions

- Page 22, rows 734-741 Should be revised to bring forth the findings and novelty of the process developed, its functionalities, characteristics and suitability.

We have reworked the whole paragraph to make it more clear what the benefits of the simulations have. (see. P.27, l.805-p.28, l.813). Also, the highlights of the technology’s functionality are emphasized within a new paragraph (see p.27, l.783-793).

We hope the addressed changes meet your approval.

Reviewer 3 Report

The following comments have been made to improve the quality of the manuscript.

1)    The title of the manuscript is long and complex, it needs revision. It should be in such a way that it should depict the exact work. A good title contains the fewest possible words that adequately describe the contents of your research work.

2) line 14-18 can be rephrased and shortened.

3)    The outcome in the abstract should be revised. Please mention the specific outcome (such as in percentage, number, temperature, etc) at the end of the abstract.

4)    The introduction is written well but the research gaps need to be highlighted.

5)    The numerical modeling methodology has been not described in detail. Could you add some governing equations, initial & boundary conditions, and model descriptions?

6)    Did you consider some assumptions? It would be good if you can add limitations of the study and supporting arguments.

7)    Which turbulence model was used in this study? What are the justifications for the choice of the model?

8)    Please discuss the importance of outputs from Fig. 10.

9)    Could you please mention the physical condition simulation?

10)        The conclusion is insufficient. Again, you should highlight the importance of particular results. Please emphasize more on strength of your research work.

11)        There are many grammatical mistakes in this work. Please make revisions for better clarity. 

12) Please check references. Some references are not as per requirements. 

Author Response

We thank the reviewer for the careful review and the helpful comments. We revised the manuscript accordingly and added the information request, especially considering our simulation work. The changes will contribute for a deeper understanding and emphases the value of the manuscript.

1)The title of the manuscript is long and complex, it needs revision. It should be in such a way that it should depict the exact work. A good title contains the fewest possible words that adequately describe the contents of your research work.

We changed the manuscript title to “Nanoval technology – an intermediate process technology between meltblown and spunbond” and hope the more compact version is more attractive. 

2) line 14-18 can be rephrased and shortened.

We comprehensively revised the abstract. Especially rows 14 to 18 have been shortened and rephrased.

3)    The outcome in the abstract should be revised. Please mention the specific outcome (such as in percentage, number, temperature, etc) at the end of the abstract.

We comprehensively revised the abstract. The specific outcome (MFI range of polymers, adjustable diameter range) was shifted from the middle of the abstract to the end to highlight it.

4)    The introduction is written well but the research gaps need to be highlighted.

We revised the Introductions section at various points also connected to comments of the other revisors. All changed passages are marked respectively. (p.2: l.53, l. 6, l.61f, l.67, l.74f, l.78f, l.83, p.3: l.99f, l.100, l.101, l.108, l. 120, p.3: l.125, l.126, l.128f, l.131, l.151-155, l.164, l.178-186

5)    The numerical modeling methodology has been not described in detail. Could you add some governing equations, initial & boundary conditions, and model descriptions?

We added the governing equations and the model description complemented with initial and boundary conditions to section 2.7 (p. 10., l.365ff).

6)    Did you consider some assumptions? It would be good if you can add limitations of the study and supporting arguments.

We made some assumptions on the modeling side. We neglect the fiber-fiber interaction and the influence of the fibers to the airflow; this is remarked on page 9, l.345. We assume the viscous material behavior near to the nozzle and visco-elastic far away. Both models are coupled at suitable point as already described in the text, page p.9,l.343ff and p.10, l.467ff. Surface tension of the polymer fibers is neglected, this is mentioned at page 9,l.348f. However, we will include this effect in the future and discuss the influence in future works.

7)    Which turbulence model was used in this study? What are the justifications for the choice of the model?

We use a shear stress transport (SST) k-ω turbulence model to restrict the computational effort compared to more complex models likes Large Eddy Simulations. The k-ω turbulence model is one of the most commonly used turbulence models and has been calibrated on a wide rage of turbulent flow data. We added this information to section 2.7, p.11. l.401f.

8)    Please discuss the importance of outputs from Fig. 10.

The figure helps to understand how the swirl is applied onto the downwards facing air stream and it shows how the process air is accelerated due to the expansion at the exit of the nozzle. The air stream conditions at the tip of the pink capillary are of special interest since they are responsible for the first forces acting on the fiber. We added this information  to the text (see. P. 19,l. 639ff).

9)    Could you please mention the physical condition simulation?

The pressure with the corresponding mass throughput and the temperature at the process air inlet have been added to the section (2-7, p.11. l.405f)

10)  The conclusion is insufficient. Again, you should highlight the importance of particular results. Please emphasize more on strength of your research work.

We revised the conclusion briefly in order to highlight the most important outcomes and strength of our study.

11)        There are many grammatical mistakes in this work. Please make revisions for better clarity.

We revised the whole manuscript. We hope we meet now a higher clarity. All changes are marked respectively in the manuscript.

12) Please check references. Some references are not as per requirements.

We checked the references and eliminated doubled citations and errors of the citing style to meet all reference requirements.

 We hope the addressed changes meet your approval.

Round 2

Reviewer 2 Report

Congratulations for your outstanding contribution.